# The role of training on smallholder farmers' adoption of orange-fleshed sweet potato in Ethiopia

Lidya Samuel[1,2]*, Marcia Dutra de Barcellos[2,3,4], Nathaline Onek Aparo[2], Mulugeta D. Watabaji[1], Hans De Steur[2]

**1** Department of Business and Economics, Haramaya University, Dire Dawa, Ethiopia, **2** Department of Agricultural Economics, Ghent University, Ghent, Belgium, **3** Department of Administrative Sciences, Federal University of Rio Grande do Sul, Porto Alegre, Rio Grande do Sul, Brazil, **4** NOVA Information Management School, Universidade NOVA de Lisboa, Lisbon, Portugal

* lidyasamuel.abayneh@ugent.be

## Abstract

Biofortified orange-fleshed sweet potato (OFSP) offers a viable strategy for combating vitamin A deficiency in Sub-Saharan Africa. Despite its nutritional benefits, its adoption among Ethiopian smallholder farmers remains limited. This study investigated the role of training in driving OFSP adoption, using a qualitative experiment grounded in the Theory of Planned Behavior and the Technology Acceptance Model. A total of 65 Ethiopian sweet potato farmers were interviewed before and after training, of which 33 were trained (17 adopters and 16 non-adopters). The findings highlight that training substantially enhanced farmers' knowledge, improving their cultivation practices and addressing key barriers such as pest control and vine availability. The training also led to a measurable improvement in OFSP yield, indicating its impact not only on behavioral intention but also on actual agronomic outcomes. Perceived ease of use and perceived usefulness emerged as important predictors of adoption behavior, with trained farmers reporting increased confidence in growing OFSP. Additionally, decisions were strongly influenced by subjective norms set by community leaders and peer farmers, underscoring the role of social dynamics in the adoption process. The dissemination of high-quality OFSP vines further supported sustained adoption. The results of this exploratory study suggest that while training is important for improving cultivation techniques, individual motivation and social support also play a key role in farmers' decision-making. Alongside the qualitative insights, quantitative yield comparisons were incorporated, and the integration of qualitative and quantitative evidence provides valuable insights into how perceptions of benefits, ease of use, and social norms shape adoption behavior, informing strategies to promote wider adoption of biofortified crops. More research is needed to further validate these findings, preferably through larger samples and longitudinal study designs.

**Data availability statement:** Due to the sensitive nature of the data collected in this study, which includes household-level socio-economic and farm management information linked to individual smallholder farmers in the Oromia region, public sharing of the full raw dataset could risk compromising participant anonymity. The data were collected under strict confidentiality agreements approved by the Ethics Committee of Haramaya University and in collaboration with the International Potato Center (CIP), both of which require that respondent identities and associated metadata remain confidential. In accordance with these institutional and ethical obligations, the full raw dataset cannot be made publicly available. However, an anonymized and aggregated dataset containing key variables essential for replicating the main findings is provided as a supplementary file to ensure transparency and reproducibility while protecting participant privacy. Data are available from the Haramaya University Ethics Committee (contact: Dr. Bobe Bedadi, email: bobedadi2009@gmail.com) for researchers who meet the criteria for access to confidential data.

**Funding:** The work of MDB was supported by the Foundation for Science and Technology (FCT Portugal) under the projects UID/97/2025 (CEGIST), UID/04152/2025 – Centro de Investigação em Gestão de Informação (MagIC)/NOVA IMS – https://doi.org/10.54499/UID/04152/2025 (2025-01-01/2028-12-31), UID/PRR/04152/2025 – https://doi.org/10.54499/UID/PRR/04152/2025 (2025-01-01/2026-06-30), and the CEEC Programme Contract [2023.09134.CEECIND/CP2836/CT0020, https://doi.org/10.54499/2023.09134.CEECIND/CP2836/CT0020]. The funders had no role in the study design, the collection, analysis, or interpretation of data, the writing of the manuscript, or the decision to publish the results.

**Competing interests:** The authors have declared that no competing interests exist.

# 1 Introduction

Despite significant advances in economic development and scientific innovation, millions of people worldwide continue to suffer from hunger and malnutrition [1,2]. Among the essential nutrients, vitamin A is particularly important as it plays a crucial role in vision, immune function, and cell growth. Nevertheless, vitamin A deficiency (VAD) remains a major public health problem, particularly in poor and low-income countries, disproportionately affecting vulnerable populations [3–5]. In Ethiopia, for instance, VAD contributes significantly to the morbidity and mortality of preschool children and women [6,7]. Although preventive strategies such as dietary diversification, food fortification, and vitamin A supplementation have been implemented, their effectiveness is often hindered by challenges related to poverty, limited technical capacity, and inadequate infrastructure [8,9]. Therefore, innovative solutions are essential to address this persistent problem effectively [10].

In recent years, biofortification has emerged as a promising strategy for combating various nutritional deficiencies, including VAD [11]. The orange-fleshed sweet potato (OFSP), rich in beta-carotene, presents a viable and cost-effective solution for mitigating VAD. in regions such as Ethiopia [12]. Despite its potential, the widespread adoption of OFSP is hindered by several barriers, including limited access to high-quality planting material, low awareness among both farmers and consumers, and firmly established dietary preferences [13]. Addressing these constraints requires coordinated efforts to strengthen farmer education, enhance the availability of OFSP planting materials, and promote awareness of its nutritional benefits among consumers [14]. Numerous studies have looked at the factors that influence farmers' and consumers' acceptance of biofortified crops. Quantitative [15–17], qualitative [18,19], and experimental studies [14] have shown that smallholder farmers' acceptance of biofortified crops, like other agricultural innovations, is shaped by a range of socio-economic, demographic, cultural, and institutional factors. Additionally, farmers' perceptions of agronomic performance and sensory attributes of these crops play an important role in their adoption decision [18–20]. Despite this growing body of literature, there remains a notable gap in understanding the dynamics of adoption behavior, particularly with respect to non-adoption and sustained use of biofortified crops. Although past studies such as Schnurr et al., 2020 and Adekambi et al., 2020 examined the factors influencing the adoption and continued use of OFSP varieties, their reliance on cross-sectional data limits insights into the evolving nature of adoption over time. Such approaches overlook behavioral shifts, including discontinuation or reversal of adoption decisions [21].

While previous studies have emphasized the importance of nutritional information in promoting the adoption of biofortified crops [14,19,22], such efforts largely fall within the domain of nutritional intervention rather than agricultural ones. Consequently, there is limited literature examining how agronomic interventions influence farmers' intentions to adopt biofortified crops and the actual outcomes of their adoption, such as improved yields or sustained cultivation. Mwiti et al., 2020, for example, argued that scaling up the cultivation of OFSP needs more than just the provision of planting material and nutritional information; it also requires overcoming information

barriers through agronomic training. Agricultural training interventions are widely recognized as pivotal for enhancing farm productivity, knowledge and fostering behavioral shifts toward sustainable practices [23–25]. Schnurr et al. (2020), for instance, demonstrated that agricultural training in Indonesia significantly improved farmers' technical competencies, facilitated the adoption of practices such as soil management and crop diversification, and ultimately contributed to increased yields and environmental sustainability.

To address the aforementioned knowledge gaps, this study aims to (1) examine the role of agronomic training in influencing farmers' intention to (continue to) adopt biofortified crops, (2) explore how such training affects the drivers and barriers influencing smallholder farmers' decisions to adopt or continue adopting OFSP, and (3) analyze the effect of agronomic training on OFSP yield among Ethiopian farmers. The study employs a qualitative methodology combined with a field experiment, referred to as a "qualitative experiment" [26–28]. In-depth interviews were conducted with farmers who either received or did not receive training on OFSP-related agronomic practices. This training intervention allowed researchers to observe differences in farmers' attitudes and perceptions, providing deeper insights into how various factors influence adoption decisions [26]. The study is guided by the following research questions: (1) What role does agronomic training play in farmers' decisions to cultivate OFSP? (2) How does training influence the key drivers of smallholder farmers' OFSP adoption? (3) What is the effect of agronomic training on OFSP yields among Ethiopian smallholder farmers?

To answer these questions, the study integrates the Technology Acceptance Model (TAM) and the Theory of Planned Behavior (TPB) as its underlying theoretical framework. The TAM is employed to examine how perceived usefulness and ease of use influence the adoption of OFSP, while the TPB provides insights into how subjective norms, perceived behavioral control, and attitudes shape farmers' intention to adopt. Combining these models offers a comprehensive understanding of the psychological and social determinants of adoption behavior. The findings from this study will provide valuable contributions to both research and policymaking, particularly in the domains of agriculture and public health. They hold relevance for key stakeholders, including ministries of agriculture, research institutions, and non-governmental organizations (NGOs) involved in the promotion of biofortified crops. By identifying critical enablers and barriers to adoption, this study aims to inform strategic interventions that promote the sustainable and continued use of biofortified crops, such as OFSP.

## 1.1 Theoretical framework

In examining the factors that influence farmers' intention to adopt agricultural innovations, previous studies have frequently applied the TPB and the TAM. The TAM, developed by Davis (1989) [29], posits that individuals' intention to adopt a technology is shaped primarily by two factors: perceived usefulness (PU) and perceived ease of use (PEOU). According to this model, technologies perceived as beneficial and easy to use are more likely to be adopted. The TPB, formulated by Davis (1989) [29], adds a psychological lens by identifying three key predictors of behavioral intention: attitudes towards the behavior, subjective norms (SN)—the perceived social pressure to perform or not perform the behavior—and perceived behavioral control—the perceived ease or difficulty of performing the behavior [30].

Several studies have validated the relevance of the TPB and TAM in agricultural settings. For instance, Sulaiman et al (2020) used the TPB to assess how subjective norms and perceived behavioral control influence farmers' adoption intentions for soil and water conservation practices in Iran, while Bagheri and Teymouri (2022) [31] explored agronomic iodine biofortification adoption in Uganda. Similarly, Aparo et al.2023 [32] applied the TAM to examine farmers' intentions to adopt biofortified rice in the Philippines. And while Ambong and Paulino (2020) [33] showed that perceived usefulness (PU) significantly influenced Nigerian farmers' willingness to adopt biofortified cassava, low perceived ease of use (PEOU) was identified as a barrier in the study [34].

Building on these insights, this study integrates the TPB and TAM frameworks to create a framework for investigating farmers' adoption and continued use of OFSP, with particular attention to the influence of agronomic training. This model captures both behavioral factors (attitudes, SN, and control perceptions) and technology-specific factors (PU and PEOU), allowing for a more comprehensive analysis of the cognitive and contextual factors driving adoption decisions. Although

widely used, each model has limitations when applied in isolation. The TPB often neglects emotional or affective influences, while the TAM tends to overlook broader contextual factors such as market access or policy incentives [35,36]. These limitations are particularly relevant in the context of OFSP, where adoption decisions are shaped by both psychological perceptions and systemic constraints that can play a role. By integrating both TPB and TAM, this study contributes a comprehensive framework Mayanja et al., 2024 [36]; Wang et al., 2023 [37] for understanding how smallholder farmers in Ethiopia respond to training interventions and make adoption decisions regarding OFSP. This approach is grounded in previous research showing that behavioral outcomes are influenced by a combination of cognitive judgments, external influences, and anticipated consequences [36,38]. Fig 1 presents the integrated theoretical model, adapted from [39], illustrating the linkages between training, technology perceptions, and behavioral intention.

While TPB and TAM are traditionally applied in quantitative research, this study innovatively applies them within a qualitative experimental design. This approach enables a more in-depth examination of the social and psychological factors influencing farmers' adoption of OFSP. Furthermore, the novelty of this study lies in its application of the integrated framework to the context of biofortification. As OFSP uptake in our study location in Ethiopia remains low despite its well-documented benefits, the study uniquely explores the interplay between farmers' behavioral intentions and external barriers such as social norms and institutional support.

## 2 Research methodology

### 2.1 Data collection procedures

As malnutrition, particularly micronutrient deficiencies such as vitamin A [8], is most prevalent in the Eastern Oromia region [40], this region was selected as the target area for this study. Eastern Oromia is primarily rural, home to many

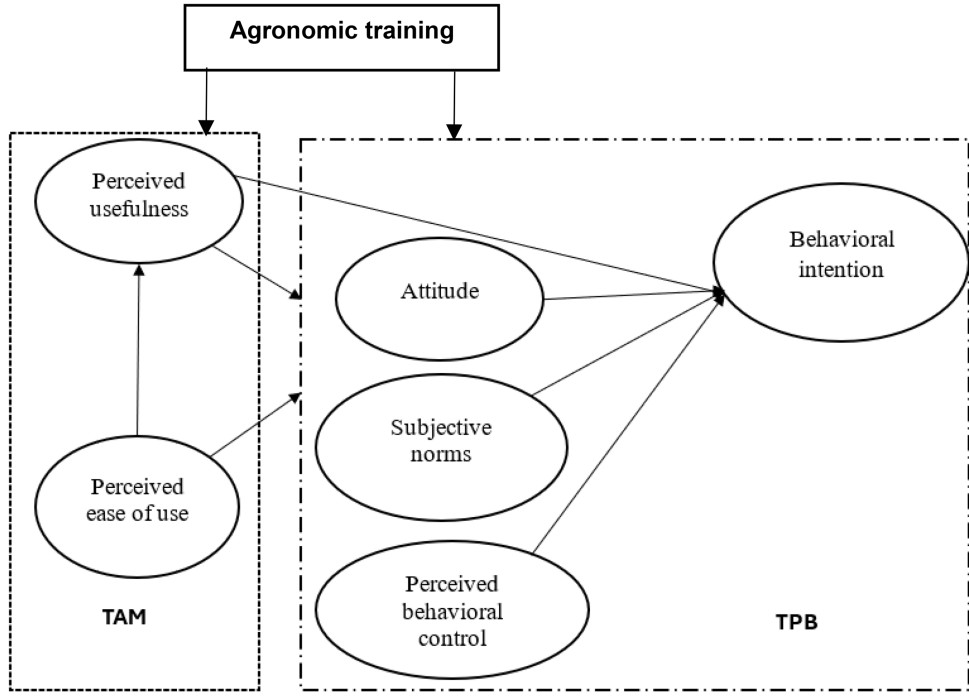

**Fig 1. The role of training on smallholder farmers' adoption of OFSP in Ethiopia: an integrated model: An Integrated Model of Theory of Planned Behavior (TPB) and Technology Acceptance Model (TAM) (adopted from Tang & Jiang, 2024).**

impoverished smallholder farmers, and is the second-largest producer of sweet potatoes in Ethiopia [41,42]. A purposive sample of 65 sweet potato farmers was selected in collaboration with the district administration and Haramaya University Research Institute (HURI). Farmers were eligible if they were 18 years or older and owned farmland.

Data collection occurred in two phases: an initial baseline study in January 2024, followed by an intervention and a post-intervention study in May 2024. The intervention started after farmers confirmed their willingness to participate in the second phase. While all 65 farmers received free OFSP vines (donated by the International Potato Center—CIP), only 33 farmers were trained in agronomic practices by CIP field research officers. A follow-up visit was conducted four months after the initial visit to assess the influence of the training intervention on yield changes and other key outcomes. Data was collected using an interview guide featuring structured and open-ended questions designed to capture the experiences of farmers from one district (Haramaya) in Tinike village.

## 2.2 Study design and experimental procedures

A qualitative experimental approach, complemented by in-depth interviews, was utilized to gain insights into farmers' attitudes, perceptions, and behaviors [26,43]. The study employed a 2x2 factorial experimental design to examine the combined effects of two factors: training and adoption (see Table 1). All farmers in the study received vines; however, the experimental intervention involved providing subsequent training to two subgroups of participants to assess the knowledge, agronomic (yield), and behavioral outcomes after the intervention. Of the 65 farmers included, 33 were randomly assigned to the training/treatment group, which consisted of 17 existing OFSP adopters and 16 non-adopters. The remaining 32 farmers, who received no training, comprised the control group, consisting of 19 adopters and 13 non-adopters. The underlying assumption of the study is that agronomic training would positively influence farmers' intention to (continue to) adopt OFSP cultivation in the next planting season and that it would also affect the yield.

The sample size in this study was unbalanced due to the nature of the sampling frame, which was not specifically constructed for this research. Instead, it was based on a list provided by the district administration and Haramaya University, comprising farmers already participating in various crop introduction projects. As a result, the number of eligible participants varied. We prioritized those farmers with accessible farmland located in secure areas to ensure logistical feasibility for follow-up monitoring and to mitigate risks associated with regional security. All farmers received OFSP vines of the same origin and variety (Dilla), a standard practice to maintain consistency in planting material quality, reduce potential variability in results, and ensure that any differences in adoption or cultivation outcomes and yield outcomes were not attributable to variations in vine quality or source. The training intervention consisted of a one-day, practical session conducted at a model farmer's field to ensure hands-on learning. A single instructor conducted all sessions to maintain instructional consistency. The training covered key agronomic practices relevant to OFSP cultivation, including land preparation, planting and multiplication techniques, pest and disease management, harvesting, and storage. The session incorporated interactive discussions and field demonstrations, reflecting best practices identified in the literature [44] and aligning with CIP's farmer training manual to ensure compliance with recommended agronomic standards and enhance the relevance and effectiveness of the intervention. The training content was specifically designed to close knowledge

**Table 1. Experimental groups.**

| Group | Category | Training | Adoption status (Baseline) | Number of farmers |
|---|---|---|---|---|
| Intervention | Group 1: Trained+Adopter | Yes | Yes | 17 |
| | Group 2: Trained+non-adopter | Yes | No | 16 |
| Control | Group 3: Not trained+Adopter | No | Yes | 19 |
| | Group 4: Not trained+non-adopter | No | No | 13 |

gaps and equip farmers with the practical skills needed for successful OFSP cultivation. The study's dependent variable was adoption intention, measured for both non-adopters and adopters.

## 2.3 Interview guide

To assess changes in farmers' adoption intentions, a pre- and post-intervention study was developed with an experimental structure adapted from the experimental methodology of De Brauw et al. (2018) [13] on OFSP adoption in East Africa. The interview guide was developed based on relevant literature and established theories linked to technology adoption (i.e., TPB-TAM) [30,35,45,46]. Like Kendall et al. (2021) [47], this study protocol was also underpinned by exploratory qualitative research conducted by the authors with both farmers and senior government officials from the districts, aimed at developing a nuanced understanding of the contextual issues.

The interview guide included four sections. The first section (Demographic and farm characteristics) collected background information on respondents, including age, gender, marital status, education level, farming experience, farm size, membership in a cooperative, access to extension services, and satisfaction with income. To respect cultural sensitivity around financial matters, farm income satisfaction was measured using a five-point Likert scale ranging from "very dissatisfied" to "very satisfied." The second section (Knowledge and experience with OFSP) explored both adopters' and non-adopters' prior exposure to OFSP, including their understanding of its nutritional benefits, previous cultivation attempts, and perceived challenges. The variables in this section were adapted from previous studies conducted in similar smallholder contexts [30,48–51]. The third section (TAM constructs) focused on farmers' perceptions of OFSP in terms of PU (e.g., nutritional or economic benefits) and PEOU (e.g., difficulty of cultivation or learning). These constructs were critical for assessing how farmers evaluated OFSP. Finally, the fourth section (TPB constructs) included questions related to farmers' attitudes toward OFSP, SN (e.g., influence of peers, family, or community leaders), and perceived behavioral control (e.g., availability of resources or knowledge needed for cultivation). These items aimed to capture the broader social and psychological drivers behind adoption intentions. To ensure cultural sensitivity and clear communication, a local coordinator fluent in Afan Oromifa was employed to support the data collection. Given the low level of education of the households, data was collected by enumerators who were fluent in Afan Oromifa and familiar with in-depth interviews in the local language from a total of 65 farmers.

For the follow-up data collection, the same interview guide was administered to 59 out of 65 farmers, incorporating additional sections to assess agronomic, behavioral, and knowledge-related outcomes. However, for the 6 farmers who chose not to plant the vines, a separate interview guide was developed to specifically explore their reasons for non-adoption. Each interview lasted approximately one hour, and all interviews were recorded with the participants' verbal consent, ensuring thorough data collection [52]. The study adhered to the principles outlined in the Declaration of Helsinki. It was approved by the Research Ethics and Integrity Committee of the Africa Centre of Excellence (ACE) for Climate Smart Agriculture and Biodiversity Conservation (Reference Number AGE 01/11/2024). Given that many participants were smallholder farmers with limited literacy, the Committee approved verbal informed consent. Prior to each interview, the study objectives, voluntary participation, and confidentiality assurances were clearly explained in the local language. Each participant's verbal consent was recorded in the enumerators' field log and witnessed by a trained field assistant. This procedure was reviewed and approved by the Ethics Committee as part of the IRB protocol.

## 2.4 Data analysis

The analysis for this study was conducted in two phases to comprehensively assess the role of agronomic training on farmers' intention to adopt (continue) adoption of OFSP, the drivers of (continued) adoption, and yields. A specific methodological approach was used in each phase, tailored to the nature of the data and the research objectives. In the qualitative phase, the focus was on carefully transcribing interview recordings using a standardized template developed by the research team to ensure consistency and reliability in capturing farmers' perspectives on OFSP adoption.

**2.4.1 Phase 1 (Qualitative analysis of baseline and endline interview data).** As recommended by Jenkins et al. (2018) [18], a detailed field log was used to document and track interactions with farmers, record personal market observations, plan interviews, and compile lists of recurring research topics. The key informant interviews (KIIs) were then transcribed and coded using MAXQDA 24, which supported a thematic, deductive approach [53]. An inductive coding process was then conducted, analyzing each transcript line by line to generate focused codes and develop a systematic category framework. Using this rigorous qualitative approach, two main theoretical themes (TAM, TPB) and 3,632 different codes were identified, ensuring a nuanced understanding of the baseline and endline data and enhancing the depth of the study. The initial codes were systematically reviewed, refined, and clustered into approximately 300 refined codes, grouped under the main theme constructs.

**2.4.2 Phase 2 (Quantitative analysis of pre-post-intervention data).** Both pre- and post-intervention data on overall yield and intention to adopt were analyzed using a quantitative approach. To assess the effect of training on OFSP yield among adopter farmers, a two-way mixed analysis of variance (ANOVA) was performed using SPSS. This technique allows us to examine the interaction between training status and prior intention to adopt on OFSP adoption behavior. A similar strategy has been applied in previous studies to assess the impact of interventions on agricultural productivity [54,55]. In this study, experimental conditions (trained vs. not-trained) and adoption status (adopter vs. non-adopter) were considered fixed factors. As such, we assessed the main effects, i.e., the direct effect of training (trained vs. not-trained) and adoption status (adopter vs. non-adopter) on OFSP yield, as well as the interaction effects, i.e., how the combination of training and adoption status influenced adoption behavior and yield. To identify specific group differences, Tukey's post hoc test was conducted for pairwise comparisons. Before conducting the ANOVA, key assumptions were verified. Normality of residuals was assessed using the Shapiro–Wilk and Kolmogorov–Smirnov tests, which indicated no significant deviations from normality ($p > 0.05$). Homogeneity of variances was tested using Levene's test, which confirmed equal variance across groups ($p > 0.05$) [56]. These results validated the use of ANOVA for the analysis. Therefore, the test results presented in the results section are based on data that met the underlying statistical assumptions. This integrated approach, combining qualitative and quantitative insights, provided a comprehensive understanding of how training influenced OFSP adoption (intention) and OFSP yield of smallholder farmers in Ethiopia.

## 3 Results and discussion

### 3.1 Farmers' characteristics

The main characteristics of the samples are presented in S1 Table. Most of the respondents from both categories were male, married household heads with an average of two children under five. The average formal schooling was five years, indicating higher illiteracy rates among Ethiopian farmers. The majority of farmers were older, with smaller farm sizes than the regional average, which is consistent with previous studies in the Oromia region [57]. They were members of cooperatives Rare Hora and Kune Mara for 11–13 years. Farmers reported an average satisfaction of 3.7 with their household income, consistent with a study by (Headey et al., 2014) [58] in Haramaya district, which also found a medium level of farmer satisfaction with income.

**3.1.1 Experience with OFSP.** In the study area, farmers were already familiar with cultivating white-fleshed sweet potato varieties before OFSP was introduced. These conventional varieties were mainly grown for household consumption and as a supplementary food source, especially during lean seasons. This prior experience with sweet potato cultivation provided a useful reference for farmers when evaluating OFSP, helping them compare its agronomic performance, taste, and market value with traditional varieties.

Pre-intervention farmers' engagement with OFSP varied substantially. Some had never cultivated it (n = 29), citing priorities for cash crops, a lack of planting material, or the absence of formal agronomic training. Others learned informally, such as by observing neighboring farms or relying on traditional knowledge. A few received nutrition-related training from initiatives like the Train-the-Trainer program from Haramaya University, the regional government, covering the process

from planting to harvesting. Extension services and community knowledge-sharing played a significant role in disseminating effective cultivation practices, including pest control and harvesting methods. A male respondent aged 34 noted: *"Yes, I have grown OFSP on my farm in the past. Although I have not received any formal training, I have gained practical experience and knowledge by interacting with other experienced farmers in my community"*. The experiences of the farmers in this study are consistent with the results of previous research on the adoption of biofortified crops. Studies have shown that access to high-quality planting material and effective extension services are critical factors for adoption. Bouis & Saltzman (2017) [59], for example, it has been highlighted that biofortification efforts need to be supported by robust seed distribution systems and farmer training programs to ensure successful adoption.

### 3.1.2 Consumption.
Prior to the intervention, farmers who consumed OFSP appreciated its flavor, nutritional benefits, and ease of preparation. Many reported eating it regularly due to its early maturity, food security benefits, and dietary variety. A 33-year-old female farmer noted: *"We eat OFSP several times a week for its taste and health benefits"*, while another highlighted its reliability during food shortages. However, non-adopters cited unfamiliarity, lack of traditional integration, and limited knowledge about preparation methods. Some farmers remained reliant on staple crops like maize and beans, indicating a need for more awareness of OFSP's dietary versatility. These responses align with studies that emphasize cultural acceptance and awareness as key factors in the adoption of biofortified crops [60,61].

### 3.1.3 Barriers and drivers to OFSP adoption.
Prior to the intervention, the study revealed that peer pressure within the farming community played a significant role in influencing adoption (intention), as summarized in Table 2. Both adopters and non-adopters observed others successfully cultivating OFSP, which encouraged them to plant the vines, fostering a sense of community expectation and shared learning. This aligns with previous studies highlighting the role of social influence in agricultural technology adoption [62]. In addition, their children's enthusiasm and fondness for the crop motivated several farmers, emphasizing the importance of growing a crop that their families like and find useful, as noted in previous research [61]. The economic potential and nutritional benefits of OFSP were also important motivating factors [63,64], especially when the availability of free vines reduced the initial investment costs, reducing financial barriers to adoption, which aligns with findings from previous studies on input subsidies and technology uptake in smallholder farming [65].

Despite the recognized benefits of OFSP, several barriers hinder its widespread adoption among both OFSP adopters and non-adopters. Most farmers reported difficulties in obtaining OFSP vines, which are primarily sourced from research institutions, universities, or neighboring farms. The lack of high-quality planting material restricts new farmers from trying the crop and limits the expansion of existing cultivation. While some OFSP adopters gained knowledge through extension services or informal networks, many lacked formal training in OFSP cultivation. The absence of structured guidance on planting techniques, pest control, and post-harvest handling posed challenges, making farmers hesitant to adopt the crop. Some OFSP adopters prioritized high-value cash crops over OFSP, viewing them as more profitable and marketable. The

**Table 2. Perceived barriers and drivers of farmers' intention to adopt OFSP.**

| Key barriers | Key drivers |
| --- | --- |
| Limited access to high-quality vines | Peer pressure and social influence within farming communities |
| Lack of formal agronomic training and guidance | Children's preference and household appreciation for OFSP |
| A preference for more profitable, high value cash crops | Nutritional benefits (e.g., high vitamin A content) |
| Perception of OFSP as a subsistence crop rather than a commercial crop | Economic potential of OFSP |
| Pest infestations and the need for intensive early-stage management | Reduced initial investment due to free vines |
| Weak market linkages and difficulties in finding consistent buyers | Knowledge dissemination through informal networks and extension agents |
| Perishability of the crop and lack of efficient storage solutions | Cost efficiency and drought resistance |
| Lack of traditional integration into local diets and limited knowledge of preparation methods | Good taste, enhancing both household consumption and market value |

perception that OFSP is primarily for household consumption rather than commercial sale reduced its appeal to farmers seeking immediate economic returns. OFSP adopters identified pest infestations and the need for careful early-stage management as obstacles to successful OFSP cultivation. Unlike traditional staple crops, with which adopters are more familiar, managing OFSP requires additional effort, thereby discouraging adoption. Farmers also faced challenges in selling OFSP due to limited market linkages and fluctuating demand. Some reported difficulties in finding buyers, while others struggled with the perishability of the crop, which necessitated timely marketing and efficient storage solutions. Non-adopters cited unfamiliarity with OFSP and a lack of traditional integration into local diets as barriers to consumption. Some farmers and consumers preferred staple crops such as maize and beans, which are deeply embedded in their food culture. Limited knowledge of preparation methods further reduced interest in OFSP adoption. These findings align with previous research on biofortified crop adoption, which emphasizes the importance of robust seed distribution systems, extension services, and community networks [62,64]. Studies in Mozambique and Tanzania similarly found that nutritional benefits, yield stability, and economic potential drive OFSP adoption [15,60,65,66]. As noted by Sheoran et al. (2022) [67] and Mwiti et al. (2020) [14], sustainable adoption of OFSP depends on quality inputs and support. As Sulaiman et al. (2020) and Samuel et al. (2025) suggest, collaboration between stakeholders is critical to overcoming these barriers and positioning OFSP as a key crop for improving food security and nutrition in similar regions.

### 3.2 The influence of training on drivers and barriers of farmers' decision to (continue to) adopt OFSP

This section examines the role of agronomic training in shaping key drivers and barriers to both the initial and continued adoption of OFSP. Drawing on qualitative insights, it highlights how training has influenced the key drivers and barriers of farmers' decisions. The analysis is supported by frequency data presented in S2 Table, which details the factors most commonly cited by participants.

**3.2.1 Knowledge.** Pre-intervention farmers recognized the nutritional and agronomic benefits of OFSP, particularly its high vitamin A content, economic advantages, and cost efficiency. Knowledge about OFSP was primarily disseminated through fellow farmers, agricultural extension workers, university staff, and various media channels, including radio. Extension agents played a vital role in introducing OFSP; some farmers learned about it through formal programs, while others received information informally through community interactions. The resilience of OFSP, characterized by its adaptability to various soil and climate conditions, as well as its low input requirements, was often cited as a reason for interest in the crop. This perception aligns with existing literature that emphasizes OFSP's resilience to drought, low fertilizer requirements, and minimal need for intensive farming methods, ultimately reducing production costs and labor demands [67]. Additionally, the crop's good flavor was frequently cited as a factor enhancing its appeal for both household consumption and marketability, confirming earlier findings by (Laurie et al., 2018) [68]. Despite these advantages, some farmers had limited familiarity with OFSP, underscoring the need for more comprehensive awareness-raising interventions. Moreover, the significance of indigenous knowledge transfer and community networks in agricultural innovation is well-documented. [69] emphasized in his theory of the Diffusion of Innovations (DOI) that interpersonal networks are essential for the spread of new agricultural practices. The fact that farmers in this study relied on traditional knowledge and community interactions supports this theory, suggesting that enhancing these informal networks could promote greater adoption of OFSP. After the intervention, trained farmers perceived they had a better understanding of proper cultivation techniques, including optimal planting times and methods for OFSP. They stated to have gained knowledge on pest and disease control, improving soil fertility, and the effective use of fertilizers. This aligns with previous studies that emphasize the importance of training and knowledge dissemination in improving agricultural practices and productivity [70,71].

**3.2.2 Perceived ease of use.** Before the intervention, both the adopters and non-adopters considered certain aspects of OFSP cultivation to be manageable, citing good land suitability, low labor requirements, and simple preparation methods as advantages. After the intervention, the PEOU changed among trained adopters and non-adopters. These

farmers highlighted access to agronomic training and trusted sources of vines, the ease of production and propagation of the crop, and its resilience to environmental conditions as key enablers. Trained farmers perceived that the training provided practical skills to overcome previously challenging aspects of OFSP cultivation, which is consistent with the findings in past literature Devaux et al. (2018) and Mwangi and Kariuki (2015) [72]. Although some challenges remained, such as the lack of local vine propagation centers, limited irrigation, and land constraints to larger-scale production, farmers perceived that the intervention established the foundation for improved adoption and better yields, supporting the conclusions of Feder et al. (1985). In contrast, untrained adopters and non-adopters primarily associated ease of use with factors such as cost-effectiveness, perceived nutritional benefits (awareness motivating adoption despite limited technical knowledge), and cultural acceptability. Although they received free vines, they continued to struggle with technical knowledge gaps and limited access to planting material. Their perceptions of simplicity continued to focus on the availability of inputs rather than improved cultivation practices, indicating limited understanding of agronomic techniques among untrained farmers. These findings support the existing literature, which emphasizes the need to combine input provision with agronomic training to ensure effective and sustainable adoption [73].

### 3.2.3 Perceived usefulness.

Before the intervention, both OFSP adopters and non-adopters recognized several key attributes of OFSP, including its high nutritional quality (particularly the beta-carotene content), pleasant taste, vibrant color, and early maturity, which enables multiple harvests within a season. These characteristics were valued for their potential to enhance food security and address malnutrition. This aligns with previous research that emphasizes the significance of high-yield potential, nutritional value, and adaptability in crop selection, as these attributes support food security, economic growth, and sustainable agricultural practices [15,16,18,19,51]. After the intervention, farmers' practices and perceptions improved. Trained farmers began to associate OFSP more strongly with concrete yield improvements, economic returns, and broader nutritional benefits. This shift reflects a deeper and more practical appreciation of the crop's value, as training equipped farmers with agronomic knowledge that translated perceived crop potential into observed or anticipated outcomes. This positive change in perception is consistent with studies that emphasize the value of agricultural education in increasing productivity and market access [74–76]. In contrast, the untrained farmers' perceptions remained unchanged, focused on the crop's inherent characteristics (e.g., yield and nutrition) rather than enhanced practices or market access. This suggests that training was instrumental not only in reinforcing existing perceptions but also in expanding farmers' understanding of OFSP's broader utility and performance in real farming conditions, confirming the role of training as a driver of both perception change and potential behavior change.

### 3.2.4 Attitude toward OFSP and OFSP adoption.

Prior to the intervention, farmers' attitudes toward adopting OFSP, particularly among non-adopters, were mainly influenced by external influences, such as community perceptions and peer acceptance, rather than direct experience with the crop. However, in line with literature [30], these attitudes were often not consistent with adoption behavior, as socially entrenched resistance to new crops required a critical mass of community members who accepted or expected the cultivation of OFSP for attitudes to effectively predict adoption. The adopters associated OFSP's bright color with nutritional benefits, particularly for eyesight and well-being, and expressed optimism about OFSP's potential to improve household nutrition, consistent with findings [14]. After the intervention, both trained adopters and non-adopters perceived OFSP as a valuable economic opportunity, reporting improved attitudes toward its ease of cultivation, short growing cycle, and adaptability. These findings align with findings from previous research [17,21].

### 3.2.5 Subjective norms.

Prior to the training, both adopters and non-adopters reported that their peers favored OFSP cultivation primarily due to its nutritional benefits and potential to increase income. However, social approval was limited by barriers such as high initial investment costs, limited access to information, and land constraints, which hindered uptake.

After training, subjective norms remained strongly positive, and all the trained farmers continued to perceive other peer farmers' OFSP acceptance as driven by economic, agronomic, and nutritional benefits. This suggests that training not only validated positive social norms but also amplified them by providing tangible results. In contrast, not-trained non-adopters

still perceived that their peers struggled with pest and disease management, inconsistent vine access, and market issues, indicating that the lack of training limited their exposure to successful examples or solutions. Similarly, trained adopters identified technical knowledge gaps as an ongoing perceived barrier among their peers, emphasizing that training helped them recognize and address challenges that others still face. Such patterns are also reflected in the work of Rasanjali et al., 2021, who observed that as farmers gain knowledge and resources often through structured interventions like training, their decision-making becomes more attuned to both social perceptions and practical limitations. Subjective norms, as emphasized in the TPB framework, significantly influenced farmers' decisions to adopt OFSP. Social influences from family, peers, and community discussions can indeed play a vital role in shaping attitudes by highlighting the nutritional and market benefits of OFSP [30].

**3.2.6 Perceived behavioral control.** Before the intervention, OFSP adopters acknowledged gaps in their knowledge of OFSP cultivation, while non-adopters expressed uncertainty about their ability to tackle key challenges, such as pest control and limited access to vines. The confidence of non-adopters in adopting OFSP was primarily influenced by the observed benefits and success stories of early adopters, which helped establish positive social norms. These correlations are also reflected in the work of Mwanga & Ssemakula (2011), who noted that this optimism was not always matched by a practical willingness to adopt.

After the intervention, all trained farmers reported a significant boost in their confidence in successfully growing OFSP. Farmers felt that the training enhanced their technical knowledge, improved their problem-solving skills, and provided clear explanations of best practices in pest management. They believed that the training empowered them to take control of key production variables, reinforcing their confidence in their ability to maintain OFSP cultivation. These findings align with previous research, which underscores the significance of continuous learning, risk-taking, and institutional support in strengthening perceived behavioral control and facilitating long-term adoption [35,77–79]. To provide a concise overview of how the agronomic training influenced key behavioral determinants, Table 3 summarizes the main pre- and post-training differences across the TAM and TPB constructs covered in the above sections.

## 3.3 The role of training on farmers' intention to (continue to) adopt OFSP, and on OFSP yields

**3.3.1 The role of training on behavioral intention.** Before the intervention, non-adopters expressed their intention to adopt OFSP due to its nutritional value, affordability, and yield. Meanwhile, OFSP adopters indicated their commitment to continue their adoption, driven by personal factors, such as knowledge about the crop, experience, and confidence in OFSP cultivation. This aligns with the farmer study on OFSP by Adekambi et al., 2020. Of the 33 farmers trained, 17 were already adopters before the intervention, while the remaining 16 were non-adopters. After receiving agronomic training as part of this study, all 16 trained non-adopters began cultivating OFSP, highlighting the critical role of comprehensive agronomic training in influencing farmers' adoption behavior. Trained OFSP adopters and non-adopters exhibited more proactive participation in OFSP cultivation. They actively sought information, accessed agricultural extension services, and participated in knowledge-sharing farmers' networks, demonstrating greater confidence in managing cultivation challenges. These behavioral changes indicate a more profound commitment to improved agricultural practices, aligning with previous findings that link farmer engagement to better agricultural outcomes [15]. For trained OFSP adopters, the intention to continue OFSP adoption was motivated by increased yield, past experience with producing the crop, and its nutritional value. For trained non-adopters, the willingness to adopt stemmed from market demand for the crop and success stories from trained adopters.

Extension workers' emphasis on the productivity and market value of OFSP further influenced the adoption decisions of trained farmers. Positive experiences with other improved root crops also motivated some non-adopters to experiment with OFSP. Moreover, agronomic training enabled one trained farmer to independently multiply and distribute vines to nine additional farmers outside the original sample. This initiative not only promotes community collaboration but also helps mitigate the risk of vine theft by increasing local access to resources.

Table 3. Summary of key pre- and post-training differences in TAM and TPB constructs.

| Construct | Pre-training perceptions | Post-training changes (after agronomic training) |
|---|---|---|
| **Perceived Usefulness (PU)** | Farmers valued OFSP for its nutritional content and early maturity, but had limited awareness of yield potential and market benefits. | Trained farmers associated OFSP with higher yield, better quality, and stronger market potential. |
| **Perceived Ease of Use (PEOU)** | Cultivation is seen as manageable but hindered by limited agronomic skills and vine access. | Trained farmers reported improved technical knowledge, better pest/disease management, and higher confidence in cultivation. |
| **Attitude toward OFSP** | Positive but passive; many viewed OFSP as a subsistence crop. | Trained farmers developed more favorable attitudes, recognizing OFSP's commercial and nutritional potential. |
| **Subjective Norms (SN)** | Peer influence and community perception encouraged interest but not consistent adoption. | Training somewhat strengthened positive social norms and validation through shared results and farmer networks. |
| **Perceived Behavioral Control (PBC)** | Farmers expressed uncertainty about their skills and resources. | Training somewhat increased confidence, technical competence, and the ability to manage production challenges. |

However, not all farmers adopted. Six farmers, including one trained adopter and five untrained non-adopters, decided not to plant the vines. Their reasons included a general dislike of the crop, viewing OFSP as unimportant, vine theft, concerns about land use efficiency, and dissatisfaction with vine quality. The trained farmer expressed concerns about efficient land use, while the untrained farmers primarily highlighted the low quality of the vines and the issue of vine theft. These varied responses highlight the impact of individual preferences, psychological and social dynamics, and economic factors on adoption behavior.

**3.3.2 The effect of training on OFSP yields.** While prior research (e.g., [18]) has acknowledged the role of training in agricultural technology adoption, this study also aims to provide empirical indications of its impact on OFSP yield. The provision of training led to significant agronomic benefits (yield) as shown by the trained farmers reporting higher yields, a better crop quality, and a better resistance to pests and diseases (see Fig 2). Before the intervention, baseline yields of OFSP adopters ranged from an average of 18.83 tons/ha for those in the not-trained group to an average 25.33 tons/

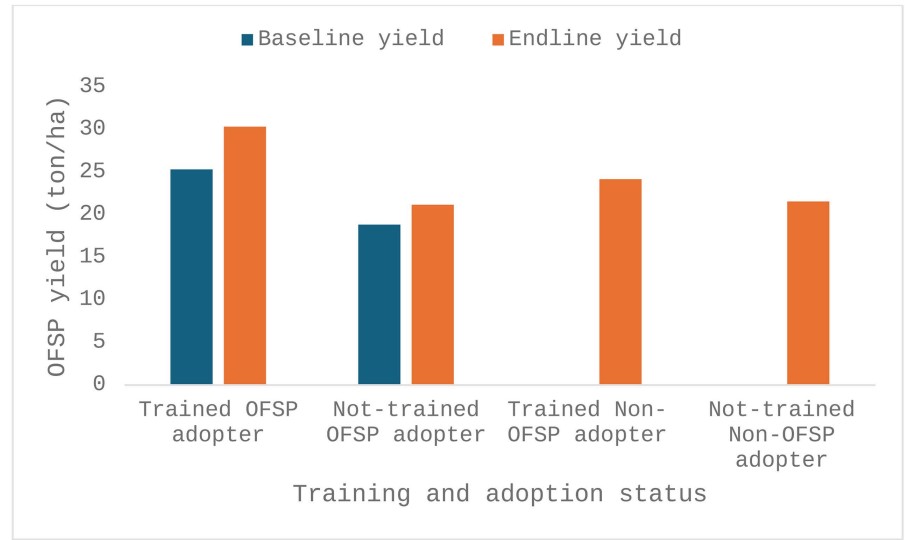

**Fig 2. Yield ton/ha between trained and not-trained OFSP adopters and non-adopters.**

ha for those in the trained group. After the intervention, trained adopters achieved a yield of 30.33 tons/ha (an average increase of 5 tons/ha), while not-trained adopters achieved 21.16 tons/ha (an average increase of 2.32 tons/ha). At the level of the non-adopters, those trained achieved 24.18 tons/ha on average, compared to an average of 21.54 tons/ha for those non-trained, showing that the training intervention resulted in yield increases for trained farmers, both adopters and non-adopters. These results are consistent with research findings indicating that access to quality inputs and training is critical to improving crop performance and productivity [80].

The results of the two-way mixed ANOVA (see Table 4) further indicate that training has a statistically significant effect on yield (F = 4.43, p = 0.039) and intention to adopt (F = 66.41, p < 0.001). In addition, a significant interaction effect was found between adoption status and training status (F = 4.93, p = 0.010), indicating that the effect of training on yield was greater for farmers with a previous OFSP adoption status, such as adopters, compared to non-adopters. In summary, the results suggest that both training and adoption intention have a statistically significant effect on yield, with a notable interaction between the two factors, as all p-values were below the 0.05 threshold.

## 3.4 Practical implications

This study demonstrates that training can stimulate (continued) OFSP adoption among smallholder farmers in Eastern Oromia, Ethiopia. Prior to the intervention, non-adopters emphasized the need for demonstration plots and training, whereas after the intervention, priorities shifted to a call for financial support and market access. Trained adopters highlighted the importance of ongoing training and vine distribution centers, whereas untrained adopters highlighted the need for community engagement and stakeholder partnerships.

Following the TAM framework, there is a need to increase the perceived usefulness, which requires showcasing OFSP's health and economic benefits through nutrition education, school feeding programs, and field trials that demonstrate yield improvements (e.g., from 25.33 tons/ha to 30.33 tons/ha for trained adopters in our study). Enhancing perceived ease of use entails improving access to extension services, quality vines, and comprehensive training. Following the TPB framework, it will be crucial to foster positive attitudes, for example, by promoting OFSP's market value and addressing misconceptions, such as the perception that it is primarily a crop for children and not profitable in the market. Leveraging subjective norms will also be necessary, which can be achieved through engaging opinion leaders and peer learning groups. Not surprisingly, peer learning and continuous feedback were especially valued by untrained adopters. Our study builds on existing literature by demonstrating how agronomic training can influence adoption barriers in agronomic knowledge-limited settings. By assessing the influence of agronomic training on farmer adoption behavior and yields, this study offers a practical approach for scaling biofortified crop programs and supporting efforts to address knowledge gaps and barriers that hinder uptake in resource-limited settings.

Our findings confirm that training in OFSP cultivation not only improves yields but also appears to affect the drivers and barriers that determine smallholder farmers' decisions to (not) adopt or continue to adopt OFSP, contributing to better economic and nutritional outcomes for farming communities. They provide actionable recommendations for policymakers and NGOs by quantifying the impacts of training, revealing local barriers (e.g., vine quality, land use), and tracking shifting

**Table 4. Two-way mixed ANOVA results of the role of training on OFSP yield of adopters and non-adopters.**

|  | Sum of Squares | df | Mean Square | F | Sig. |
|---|---|---|---|---|---|
| Adoption status | 43773.1 | 2 | 21886.6 | 66.40 | 0.000 |
| Training status | 1460.5 | 1 | 1460.5 | 4.43 | 0.039 |
| Adoption status * Training status | 3252.2 | 2 | 1626.1 | 4.93 | 0.010 |

Tests of Between-Subjects Effects with yield (ton/ha) as dependent Variable.

farmer priorities. A holistic, adaptable, and community-driven approach embedded in broader agricultural policies is essential for sustained adoption.

### 3.5 Study limitations

This study has several limitations. The small, region-specific sample in Eastern Ethiopia restricts generalizability. Future studies could improve representativeness by employing larger samples with better matching between treatment and control groups to ensure baseline comparability. Its qualitative design, while insightful, lacks quantitative validation of adoption drivers. The short time frame also captures only the initial and early adoption stages, and not the potential long-term effects of the training. Moreover, the provision of free vines may have inflated adoption rates and perceptions, potentially misrepresenting market-driven behavior. Despite the in-depth nature of our study, unaddressed external factors (e.g., market shifts, climate, and politics) and biases (e.g., interviewer bias and social desirability) may have also influenced the findings. To address the above, future research could target larger, more diverse, cross-country samples, using mixed-methods approaches with potential follow-up measures. It could also consider investigating how access to quality vines and local market dynamics influence the sustained adoption of OFSP, as well as how gender roles influence decision-making in OFSP cultivation.

## 4 Conclusions

This study offers an empirical and methodological contribution to the literature on the uptake of biofortified crops. First of all, the results confirm that training can boost farmers' intention to adopt OFSP by enhancing its perceived ease of use and usefulness, while also reinforcing positive attitudes, social norms, and perceived behavioral control. It was observed that these factors generally improved after the intervention for both adopters and non-adopters. Additionally, the study demonstrates that the training had a positive impact on actual farm performance, as evidenced by post-intervention yield data indicating an increase in OFSP yields among trained farmers. This highlights the potential dual role of training in both motivating behavioral change and improving agronomic outcomes. When integrated into broader agricultural and nutrition policies, interventions that include agronomic training and address barriers such as the need for enhanced extension services and the establishment of vine multiplication centers can potentially further increase both the uptake and sustained use of OFSP.

Secondly, by integrating TPB and TAM, this study offers a comprehensive and nuanced framework for understanding farmers' decision-making processes. This dual theoretical perspective enabled the identification of important drivers and barriers of adoption intentions. The qualitative experimental design, based on real-life conditions, provides rich evidence of how farmers' perceptions, motivations, and challenges change over time. These insights are beneficial for developing culturally appropriate, community-oriented interventions that consider the constraints and preferences of smallholder farmers.

This study provides direct evidence for policy and development efforts focused on promoting biofortified crops like OSFP, and can serve as a valuable resource for researchers, policymakers, NGOs, and agricultural extension services. They present actionable strategies to support sustainable adoption pathways for OFSP and other biofortified crops in vulnerable farming communities. Moreover, the findings contribute to the broader contexts of biofortification by showing how behaviorally informed, training-based interventions could be adapted to similar smallholder contexts across sub-Saharan Africa and other low-income regions, thus supporting the scaling of nutrition-sensitive agriculture beyond Ethiopia.

## Supporting information

**S1 Table. Socio-demographic and farm characteristics.**
(DOCX)

**S2 Table. Farmer perceptions of TAM and TPB constructs.**
(DOCX)

**S3 Data. Aggregated dataset for OFSP adoption analysis.**
(XLSX)

## Acknowledgments

We sincerely thank the International Potato Center (CIP) Ethiopian country office for their crucial support in providing OFSP vines and training the farmers. Special thanks to Mr. Sulieman Abdurhaman for his dedicated facilitation during the study. We also greatly appreciate the farmers of Tinike for their cooperation and willingness to participate in this research.

## Author contributions

**Conceptualization:** Lidya Samuel, Marcia Dutra de Barcellos.

**Data curation:** Lidya Samuel, Marcia Dutra de Barcellos.

**Formal analysis:** Lidya Samuel, Hans De Steur.

**Funding acquisition:** Lidya Samuel.

**Investigation:** Lidya Samuel, Hans De Steur.

**Methodology:** Lidya Samuel, Marcia Dutra de Barcellos.

**Project administration:** Hans De Steur.

**Resources:** Mulugeta D Watabaji.

**Software:** Lidya Samuel, Hans De Steur.

**Supervision:** Marcia Dutra de Barcellos, Mulugeta D Watabaji, Hans De Steur.

**Validation:** Marcia Dutra de Barcellos, Hans De Steur.

**Visualization:** Lidya Samuel, Marcia Dutra de Barcellos, Nathaline Onek Aparo, Hans De Steur.

**Writing – original draft:** Lidya Samuel.

**Writing – review & editing:** Lidya Samuel, Marcia Dutra de Barcellos, Nathaline Onek Aparo, Mulugeta D Watabaji, Hans De Steur.

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
