## [Decision Letter · Decision Letter 0]

29 Oct 2025

Dear Dr. Samuel,

Thank you for submitting your manuscript to PLOS ONE. After careful consideration, we feel that it has merit but does not fully meet PLOS ONE’s publication criteria as it currently stands. Therefore, we invite you to submit a revised version of the manuscript that addresses the points raised during the review process.

1. Title and Keywords – Revise the title for precision and consistency, for example: “The role of training on smallholder farmers' adoption of orange-fleshed sweetpotato in Ethiopia.” Ensure that ‘OFSP’ and ‘adoption’ are included among the keywords for indexing purposes.

2. Abstract – Briefly mention your analytical approach (integration of qualitative and quantitative data) to enhance methodological transparency.

3. Introduction – Streamline the number of early citations for readability and include a note on whether white or cream-fleshed sweetpotato was cultivated prior to the OFSP intervention in the study area.

4. Conceptual Framework and Figure 1 – Reviewer 1 observed that the linkage between training, technology perception, and behavioral intention is not fully clear in Figure 1. Please revise the figure for alignment with the text and define all constructs (e.g., PBC).

5. Methodology Clarifications –

   - Explain how random assignment to trained and untrained groups was implemented within a purposive sample.

   - Specify whether the training covered agronomic, nutritional, or both aspects.

   - Provide a statement confirming that ANOVA assumptions (normality and homogeneity) were validated.

   - Describe the scales used for rating responses and coding qualitative data.

   - Ensure that the Data Availability section conforms fully to PLOS ONE’s requirements.

6. Analytical Approach – If possible, add summary statistics or basic percentages to supplement qualitative descriptions. Reviewers also suggested acknowledging, in your discussion, that regression-based approaches (binary or ordered logistic) could be explored in future work to deepen causal inference.

7. Results and Discussion –

   - Limit direct quotations from farmers to maintain conciseness and focus on analytical synthesis.

   - Add a concise table summarizing key pre- and post-training differences in TAM/TPB constructs.

   - Consider discussing whether inclusion of food preparation and value-chain training could further influence adoption outcomes.

8. Figures and Tables – Ensure all figures and tables are correctly formatted and labeled per journal standards. Remove all placeholder text (e.g., “[INSERT FIGURE 1]”).

9. Language and Referencing – Review the manuscript carefully for grammar, punctuation, and typographic consistency. Use a single, consistent style for references and ensure journal names and titles follow PLOS ONE guidelines.

10. Conclusion – Strengthen the conclusion by briefly discussing how your findings could inform the scaling of biofortification programs beyond Ethiopia.

We commend your efforts in integrating behavioral theory with empirical agricultural research. Your study makes a valuable contribution to understanding how training interventions influence farmers’ adoption behavior and agronomic performance. We encourage you to address each point thoroughly and provide a detailed response letter explaining how revisions were made.

We look forward to receiving your revised manuscript.

Kind regards,

Julius Adewopo, Ph.D.

Academic Editor

PLOS ONE

Journal Requirements:

2. In the ethics statement in the Methods, you have specified that verbal consent was obtained. Please provide additional details regarding how this consent was documented and witnessed, and state whether this was approved by the IRB

“Ghent University Special Research Fund (BOF-01W00421)”

Please state what role the funders took in the study.  If the funders had no role, please state: "The funders had no role in study design, data collection and analysis, decision to publish, or preparation of the manuscript.”

4. Please note that funding information should not appear in the Acknowledgments section or other areas of your manuscript. We will only publish funding information present in the Funding Statement section of the online submission form. Please remove any funding-related text from the manuscript.

5. We note that you have indicated that there are restrictions to data sharing for this study. For studies involving human research participant data or other sensitive data, we encourage authors to share de-identified or anonymized data. However, when data cannot be publicly shared for ethical reasons, we allow authors to make their data sets available upon request. For information on unacceptable data access restrictions, please see http://journals.plos.org/plosone/s/data-availability#loc-unacceptable-data-access-restrictions.

6. In the online submission form, you indicated that:

“The data will only be available based on a reasonable request.”

3. Uploaded as supplementary information.

7. When completing the data availability statement of the submission form, you indicated that you will make your data available on acceptance. We strongly recommend all authors decide on a data sharing plan before acceptance, as the process can be lengthy and hold up publication timelines. Please note that, though access restrictions are acceptable now, your entire data will need to be made freely accessible if your manuscript is accepted for publication. This policy applies to all data except where public deposition would breach compliance with the protocol approved by your research ethics board. If you are unable to adhere to our open data policy, please kindly revise your statement to explain your reasoning and we will seek the editor's input on an exemption. Please be assured that, once you have provided your new statement, the assessment of your exemption will not hold up the peer review process.

8. Please include a separate caption for each figure in your manuscript.

**Additional Editor Comments:**

Dear Authors,

Thank you for submitting your manuscript to PLOS ONE. The reviewers and I appreciate your rigorous research exploring the behavioral and agronomic impacts of agronomic training on the adoption of biofortified orange-fleshed sweet potatoes (OFSP) among Ethiopian smallholder farmers. The integration of the Technology Acceptance Model (TAM) and the Theory of Planned Behavior (TPB) within a qualitative experimental design is particularly commendable and provides novel insights into behavioral change and technology adoption.

All three reviewers recognize the significance, methodological soundness, and policy relevance of your study. However, each reviewer has raised several points that must be addressed to ensure the manuscript meets PLOS ONE’s editorial and methodological standards. The decision at this stage is that this manuscript can be accepted, contingent on the revisions outlined below.

Key Revisions Required

1. Title and Keywords: Revise the title for precision and consistency, for example: “The role of training on smallholder farmers' adoption of orange-fleshed sweetpotato in Ethiopia.” Ensure that ‘OFSP’ and ‘adoption’ are included among the keywords for indexing purposes.

2. Abstract: Briefly mention your analytical approach (integration of qualitative and quantitative data) to enhance methodological transparency.

3. Introduction: Streamline the number of early citations for readability and include a note on whether white or cream-fleshed sweetpotato was cultivated prior to the OFSP intervention in the study area.

4. Conceptual Framework and Figure 1: Reviewer 1 observed that the linkage between training, technology perception, and behavioral intention is not fully clear in Figure 1. Please revise the figure for alignment with the text and define all constructs (e.g., PBC).

5. Methodology Clarifications

- Explain how random assignment to trained and untrained groups was implemented within a purposive sample.

- Specify whether the training covered agronomic, nutritional, or both aspects.

- Provide a statement confirming that ANOVA assumptions (normality and homogeneity) were validated.

- Describe the scales used for rating responses and coding qualitative data.

- Ensure that the Data Availability section conforms fully to PLOS ONE’s requirements.

6. Analytical Approach: If possible, add summary statistics or basic percentages to supplement qualitative descriptions. Reviewers also suggested acknowledging, in your discussion, that regression-based approaches (binary or ordered logistic) could be explored in future work to deepen causal inference.

7. Results and Discussion

- Limit direct quotations from farmers to maintain conciseness and focus on analytical synthesis.

- Add a concise table summarizing key pre- and post-training differences in TAM/TPB constructs.

- Consider discussing whether inclusion of food preparation and value-chain training could further influence adoption outcomes.

8. Figures and Tables: Ensure all figures and tables are correctly formatted and labeled per journal standards. Remove all placeholder text (e.g., “[INSERT FIGURE 1]”).

9. Language and Referencing: Review the manuscript carefully for grammar, punctuation, and typographic consistency. Use a single, consistent style for references and ensure journal names and titles follow PLOS ONE guidelines.

10. Conclusion: Strengthen the conclusion by briefly discussing how your findings could inform the scaling of biofortification programs beyond Ethiopia.

We commend your efforts in integrating behavioral theory with empirical agricultural research. Your study makes a valuable contribution to understanding how training interventions influence farmers’ adoption behavior and agronomic performance. We encourage you to address each point thoroughly and provide a detailed response letter explaining how revisions were made.

Reviewers' comments:

Reviewer's Responses to Questions

**Comments to the Author**

1. Is the manuscript technically sound, and do the data support the conclusions?

Reviewer #1: Partly

Reviewer #2: Partly

Reviewer #3: Yes

2. Has the statistical analysis been performed appropriately and rigorously?

Reviewer #1: I Don't Know

Reviewer #2: Yes

Reviewer #3: Yes

3. Have the authors made all data underlying the findings in their manuscript fully available?

Reviewer #1: Yes

Reviewer #2: Yes

Reviewer #3: Yes

4. Is the manuscript presented in an intelligible fashion and written in standard English?

Reviewer #1: No

Reviewer #2: Yes

Reviewer #3: Yes

Reviewer #1: My suggestions and some questions for the authors to enhance the document are outlined below.

1. I suggest reviewing the title to read “The role of training on smallholder farmers' adoption of orange-fleshed sweetpotato in Ethiopia.”

2. The manuscript needs editing to enhance language and writing quality.

3. Review the keywords and include “OFSP” and "adoption."

4. The authors should include information on whether the white/cream fleshed sweetpotato was cultivated in the studied areas before the introduction of the OFSP.

5. In Figure 1, the linkage of training to technology perceptions and behavioral intention is not apparent, as indicated in the text (page 7). Please confirm.

6. Minimise abbreviations to the least possible. When there are too many, they distract the reader as they search for meaning. What is PBC (page 11)?

7. Describe the scales employed for rating responses in the Materials and Methods section.

8. Was the training mentioned on page 13 about plant or human nutrition?

9. Although concepts and farmers’ experiences were assessed qualitatively, if any numerical data was collected, basic statistics such as percentages could be included in the results. Words like some, many, and a few do not sufficiently quantify the population segments.

10. Would the results have differed if the training had also covered food preparation and uses (covering the whole OFSP value chain)? This question may be answered in the Discussion section.

Reviewer #2: 1. Binary Logistic Regression is useful tool to appliy here, If the dependent variable “adoption” is binary (1 = adopted, 0 = not adopted). why was not used here.

2. Multinomial or Ordered Logistic Regression is also useful tool if you have multiple levels (foe example, no adoption, partial adoption, full adoption). It allows differentiation between degrees of adoption behavior.

3. You should estimates the causal impact of training on adoption by matching trained and untrained farmers who share similar characteristics.

Reviewer #3: The manuscript presents a well-structured and relevant study exploring the role of training in promoting the adoption of biofortified orange-fleshed sweet potatoes among Ethiopian smallholder farmers. The integration of TAM and TPB frameworks adds theoretical depth, and the findings have strong policy implications. Minor revisions are needed to improve methodological clarity, data availability compliance, and formatting consistency before acceptance.

**Do you want your identity to be public for this peer review?** For information about this choice, including consent withdrawal, please see our Privacy Policy

Reviewer #1: No

Reviewer #2: No

Reviewer #3: **Yes:**  Sadia Ansar

---

## [Author Response · Author response to Decision Letter 1]

25 Nov 2025

Due to the sensitive nature of the data collected in this study, which includes household-level socioeconomic and farm management information linked to individual smallholder farmers in the Oromia region, public sharing of the full raw dataset could risk compromising participant anonymity. The data were collected under strict confidentiality agreements approved by the Ethics Committee of Haramaya University and in collaboration with the International Potato Center (CIP), both of which require that respondent identities and associated metadata remain confidential.

In accordance with these institutional and ethical obligations, the full raw dataset cannot be made publicly available. However, an anonymized and aggregated dataset containing key variables essential for replicating the main findings is provided as a supplementary file to ensure transparency and reproducibility while protecting participant privacy. Data are available from the Haramaya University Ethics Committee (contact: Dr. Bobe Bedadi, email: bobedadi2009@gmail.com) for researchers who meet the criteria for access to confidential data.

---

## [Editor Report · Decision Letter 1]

26 Dec 2025

The role of training on smallholder farmers' adoption of orange-fleshed sweet potato in Ethiopia

PONE-D-25-37573R1

Dear Dr. Samuel,

We’re pleased to inform you that your manuscript has been judged scientifically suitable for publication and will be formally accepted for publication once it meets all outstanding technical requirements.

Kind regards,

Julius Adewopo, Ph.D.

Academic Editor

PLOS One
---

## [Editor Report · Acceptance letter]

PONE-D-25-37573R1

PLOS One

Dear Dr. Samuel,

I'm pleased to inform you that your manuscript has been deemed suitable for publication in PLOS One. Congratulations! Your manuscript is now being handed over to our production team.

Kind regards,

on behalf of

Dr. Julius Adewopo

Academic Editor

PLOS One